# Socio-Territorial Inequities in the French National Breast Cancer Screening Programme—A Cross-Sectional Multilevel Study

**DOI:** 10.3390/cancers13174374

**Published:** 2021-08-30

**Authors:** Quentin Rollet, Élodie Guillaume, Ludivine Launay, Guy Launoy

**Affiliations:** Centre François Baclesse, U1086 “ANTICIPE” INSERM, University of Caen Normandie, 3, Avenue du Général Harris, 14000 Caen, France; elodie.guillaume@unicaen.fr (É.G.); ludivine.launay@unicaen.fr (L.L.); guy.launoy@unicaen.fr (G.L.)

**Keywords:** early detection of cancer, breast neoplasms, organized screening programme, opportunistic screening, health-care disparities, social deprivation

## Abstract

**Simple Summary:**

France implemented a national breast cancer screening programme in 2004, which, despite recommendations, still coexists with opportunistic screening practices. We aimed to study socio-territorial inequities in participation in the 2013–2014 screening campaign, using multilevel models. With a representative sample of 42% of the estimated eligible population, we found that the organized programme does not erase social or territorial inequities in participation. Social inequities, at multiple levels, were found in nearly all *départements*, whereas territorial inequities seemed more context dependent. The impact of the coexistence with opportunistic screening, beyond any control and evaluation, is adding more risks (over-diagnosis, over-treatment) and leads to underestimating the true coverage of the population, mainly in the wealthiest, therefore leading to an underestimation of the true social gradient in participation. The French breast cancer screening programme needs to evolve to be more efficient in coverage, notably through the reduction of the unfair inequities in participation.

**Abstract:**

Background. France implemented in 2004 the French National Breast Cancer Screening Programme (FNBCSP). Despite national recommendations, this programme coexists with non-negligible opportunistic screening practices. Aim. Analyse socio-territorial inequities in the 2013–2014 FNBCSP campaign in a large sample of the eligible population. Method. Analyses were performed using three-level hierarchical generalized linear model. Level one was a 10% random sample of the eligible population in each *département* (*n* = 397,598). For each woman, age and travel time to the nearest accredited radiology centre were computed. These observations were nested within 22,250 residential areas called “Îlots Regroupés pour l’Information Statistique” (IRIS), for which the European Deprivation Index (EDI) is defined. IRIS were nested within 41 *départements*, for which opportunistic screening rates and gross domestic product based on purchasing power parity were available, deprivation and the number of radiology centres for 100,000 eligible women were computed. Results. Organized screening uptake increased with age (OR_1SD_ = 1.05 [1.04–1.06]) and decreased with travel time (OR_1SD_ = 0.94 [0.93–0.95]) and EDI (OR_1SD_ = 0.84 [0.83–0.85]). Between *départements*, organized screening uptake decreased with opportunistic screening rate (OR_1SD_ = 0.84 [0.79–0.87]) and *départements* deprivation (OR_1SD_ = 0.91 [0.88–0.96]). Association between EDI and organized screening uptake was weaker as opportunistic screening rates and as *département* deprivation increased. Heterogeneity in FNBCSP participation decreased between IRIS by 36% and between *départements* by 82%. Conclusion. FNBCSP does not erase socio-territorial inequities. The population the most at risk of dying from breast cancer is thus the less participating. More efforts are needed to improve equity.

## 1. Introduction

Worldwide, one in six deaths is due to cancer. Breast cancer (BC), despite being mostly a female disease (less than 1% occurs in men), has now surpassed lung cancer as the most frequently diagnosed cancer, all sexes combined, with 2.3 million cases in 2020. With 685,000 deaths the same year, it ranked fifth for the most common cause of cancer death all sexes combined, and first in women [1]. In France, after a stabilization between 2003 and 2010, incidence has increased again during 2010–2018. In contrast, mortality slowly decreased between 1990 and 2018. Therefore, in 2018, BC was the most diagnosed cancer (58,459 new cases) and ranked third in mortality all sexes combined, and first in women (12,146 deaths) [2].

To control BC mortality [3], France has implemented in 2004 the French National Breast Cancer Screening Programme (FNBCSP). The ambition of screening is to detect the disease at an early stage to improve its prognosis via more effective treatment. This is a national population-based programme targeted towards women aged 50–74 with medium risk of BC (no familial or personal history of the disease, no genetic disposition, and no symptoms of BC). Eligible population is invited, every two years, to perform a free of charge screening mammography and a clinical breast exam in an accredited radiology centre of her living (and sometimes bordering) *département* (an administrative division of France). Once performed, a double reading is ensured for each negative mammogram. At the time of our study, FNBCSP territorial organization was led at the *département* level by screening management structures. One of many performance indicators regarding an organized screening programme is the participation-to-invitation rate, evaluated “acceptable” at 70% by the European Commission to significantly reduce mortality [4]. According to *Santé Publique France*, the French national public health agency in charge of evaluating the FNBCSP, participation reached a peak of 52.4% in 2011–2012, then slowly decreased to 48.6% in 2018–2019 [5].

Despite the National recommendations [6], this screening programme co-exists with opportunistic screening, where mammography realization depends on medical prescription by health-care providers (mostly general practitioners or gynaecologists). It is not recommended as it does not ensure double-reading that is performed in an accredited radiology centre, is not free of charge, and is not under enough monitoring to allow sufficient evaluation.

Beyond participation, another aim of the FNBCSP is to ensure equity of access to screening [7]. Multiple publications have shown that this goal was not reached, with individual factors associated with lower participation, such as poor socioeconomic status [8,9,10,11,12], poor health [11,13] or healthcare system barriers [10,11,12,14]. Territorial inequities have also been reported, with lower screening rates among eligible women living further from the accredited radiology centres [15,16]. In recent years, attention has also increased in the association between area-based deprivation and participation. Higher deprivation was associated with lower participation in the FNBCSP in two *départements* (over 101) [15,17], and in a representative sample of the eligible population covered by the three main health insurance schemes in thirteen *départements* [16]. In metropolitan France (95 *départements*), an ecological study found a more complex relation between participation and deprivation, described as an “inverse U-curve” [18], but lacked individual data and used population estimates for outcome assessment.

In this context, we aimed to evaluate socio-territorial inequities in the FNBCSP 2013–2014 campaign by studying individual and contextual factors in a single model, in a large sample of the eligible population residing in 41 *départements* of metropolitan France.

## 2. Materials and Methods

Redaction of this article follows the Strengthening the Reporting of Observational Studies in Epidemiology guidelines [19].

### 2.1. Population and Sample

Screening management structures were invited to send their data regarding the 2013–2014 invitation campaign. Participation of the structures was on a voluntary basis. These data corresponded to the follow-up of the FNBCSP and included eligible women’s addresses of residence, dates of birth, dates of invitation (from which we computed age at invitation), and whether they performed a mammography screening through the FNBCSP in the two years following invitation. We received 4,236,066 observations nested in 41 *départements*. Data management was performed, leaving 4,001,225 (94%) unique eligible women, 42% of the estimated eligible population in France. Before geolocalisation, we performed a stratified random sampling by drawing 10% of the eligible population in each *département* (*n* = 400,125). Comparisons between samples and *départements*’ populations (data not shown) showed no important differences in participation rates (from 0 to 1.6% difference) and age (no differences exceeded ¼ years in mean age). Additional exclusions were performed due to geolocalisation and after geolocalisation. Final sample consisted of 397,598 women. Flow chart of the population is available in Figure 1.

### 2.2. Variables

Level 1—Individual level (*n* = 397,598)

Age at invitationTravel time to the nearest accredited radiology centre

We asked screening management structures which centres were active on the period. All these centres have been geocoded. The travel time to the nearest accredited radiology centre (NARC) was computed for all individuals using Navstreets^©^ V14.0 data (ESRI, 21, rue des Capucins 92195 Meudon Cedex, France; Navmart, 8400 E Crescent Pkwy Suite 652, Greenwood Village, CO 80111, United States), using the Network Analyst extension of ArcGIS^©^ PRO software (ESRI, 21, rue des Capucins 92195 Meudon Cedex, France).

Level 2—IRIS level (*n* = 22 250)

French version of the European Deprivation Index (EDI)

Thanks to geolocalisation, each woman was allocated to her residential IRIS (*Îlots Regroupés pour l’Information Statistique*), the smallest geographical unit for which census data is available. They are either neighbourhood proxies in all municipalities with more than 10,000 inhabitants and in most municipalities with more than 5000, and to the municipalities themselves in other cases. Each of these IRIS correspond to an EDI score [20], computed with the 2011 census data. Briefly, this ecological index is based on fundamental needs associated with both objective and subjective poverty, a higher EDI score reflecting higher deprivation in the area.

Level 3—*Département* level (*n* = 41)

Opportunistic screening

Opportunistic screening practices are not routinely collected at the individual level. We used the estimations from *Santé Publique France* [21], computed using the national health data system for the population of women aged 50–75 in 2013–2014. They defined these data as a “crude indicator”, because of the impossibility to differentiate opportunistic screening from mammography following clinical anomalies, or a follow-up mammogram for high-risk women. Numeric values might suffer from imprecisions; it should be interpreted as a proxy for the propensity of the population to resort to opportunistic screening procedures.

Care offer

The number of accredited radiology centres for 100,000 eligible women was calculated for each *département*.

*Département* socioeconomic level

Two indicators have been used in this study. The first one, income-based, was the gross domestic product per capita based on purchasing power parity (GDP-PPP) in 2015 according to the OECD statistics [22]. The second, deprivation-based, was the population mean of the EDI by *département*, entitled “*département* deprivation” for the rest of the document.

### 2.3. Statistics

#### 2.3.1. Centring and Standardization

All variables have been centred for interpretational reasons and standardized for computational reasons. For the rest of the document, a reference individual will be an individual for which all variables are equal to their mean, and the variable’s effect size for an increase of one standard deviation.

#### 2.3.2. Model Building

Eligible women were nested in the IRIS, themselves nested in the *départements*. To get an accurate modelling of the probability of FNBCS participation according to individuals and area characteristics, multivariate hierarchical generalized linear model was used. The statistical models were built level by level, by increasing complexity. The first model presented (Model 0) is the “null model”, a model without any explanatory variable, only composed of fixed and random intercepts. Second model (Model 1) was built by adding level 1 variables’ fixed effects and testing random slopes at higher levels. Third (Model 2) and final (Model 3) models were built on the same logical steps, with the addition of testing for cross-level interactions. Model selection was done by comparing deviance using ANOVA, and variance confidence intervals were computed using bootstrap.

#### 2.3.3. Additional Measures

To give an easier interpretation of the random intercepts variance, we computed the variance partition coefficient (VPC) [23] using the threshold latent variable hypothesis [24] and the proportional change of the variance (PCV) [25]. Random slopes models and random effects’ correlations implies that between-group variance is a function of the variables with random coefficients [26]. When a random slope is included, the variance reported in the table is the variance for a mean individual, and we plotted the more complex variation according to lower-level variables. These are quadratic functions; variance of extreme values should thus be interpreted with caution. R version 4.0.0 was used for analyses and artworks.

## 3. Results

### 3.1. Population

Description of the population is available in Table 1 and list of participating *départements*, sample sizes and participation rates are available in Appendix A. Overall participation was 55%, with marked disparities between *départements* (from 40.8% in *Essonne* and *Seine-Saint-Denis* to 68.3% in *Indre-et-Loire*). We illustrated univariate analyses of the relation between FNBCSP participation, EDI and travel time by *département* (Appendix A). Briefly, participation by population’s quintiles of the EDI followed a strong pattern, with lower participation as deprivation increased in almost all *départements*. Participation by travel time was less straightforward. It was lower among the very close population than for those who live a little further away. Then, participation seemed to decrease as distance increased.

### 3.2. Results

All model results are described in Table 2.

Model 0: There was heterogeneity in FNBCSP participation (i.e., random intercepts variance) around the fixed intercept (OR = 1.32 [1.22–1.45]) at both IRIS (σ^2^ = 0.055; VPC = 1.6%) and *département* levels (σ^2^ = 0.082; VPC = 2.4%). Shrunken residuals used to estimate these variances are illustrated in Figure 2a,b (Model 0).

Model 1: Overall, FNBCSP participation increased with age (OR = 1.05 [1.03–1.07]) and decreased with travel time (OR = 0.98 [0.96–0.99]). As shown by the random slopes, and illustrated in Figure 2c,d (Model 1), strength of these effects varied across *départements*, in such a way that the relation was insignificant or reversed in some cases. Random effects correlations showed that *départements* with higher intercepts tended to have a stronger effect for travel time and a weaker effect for age. It led to higher heterogeneity between *départements* for younger women and those closest to and furthest from the NARC (Figure 3a,b (Model 1)). There was an interaction between age and travel time (OR = 0.99 [0.98–1.00]), illustrated in Figure 4a.

Model 2: Overall, an increase in EDI was associated with lower probability of FNBCS participation (OR = 0.84 [0.82–0.86]). Accounting for EDI reduced travel time effect heterogeneity (Figure 3b (Model 2)). As shown by the random slope and illustrated in Figure 2e, strength of the association between EDI and FNBCSP participation varied across *départements*, but few had a weak relationship. Random effects correlations showed that d*épartements* with higher random intercepts tended to have a stronger effect of EDI. It led to more heterogeneity in FNBCSP participation among the wealthiest women, and, to a lesser extent, the most deprived (Figure 3c (Model 2)). Accounting for EDI also reduced random intercepts variances at IRIS and *département* levels by 34% (Figure 2a (Model 2) and 12.2%.Model 3: FNBCSP participation was lower as *départements’* opportunistic screening rates (OR = 0.84 [0.79–0.87]) and *départements’* deprivation (OR = 0.91 [0.88–0.96]) increased. There were cross-level interactions between opportunistic screening rates and both age (OR = 1.02 [1.01–1.04]) and EDI (OR = 1.04 [1.03–1.06]). As illustrated in Figure 4b,c, FNBCSP participation in *départements* with high opportunistic screening rates was lower as age and deprivation decreased. There was a cross-level interaction between *départements’* deprivation and EDI (OR = 1.02 [1.00–1.03]), with lower participation as deprivation decreased (Figure 4d). These effects reduced the remaining variance across *départements* by 79.2% (Figure 2b (Model 3)). They also strongly reduced heterogeneities between *départements* in the strength of the effects of age, travel time and EDI (Figure 2c–e (Model 3)). In addition, random effects correlations were reduced to statistical insignificance. Unexplained remaining variances between *départements* were thus independent of the lower-level variables. (Figure 3a–c (Model 3)). GDP-PPP and the number of radiology centres per 100,000 eligible women were not associated with FNBCSP participation.

## 4. Discussion

In this large sample of eligible women for the 2013–2014 screening campaign, our results strengthen the converging findings that the FNBCSP produces both territorial and social inequities in participation. Our findings about the effect of travel time are coherent with other studies led in France [15,16], while results are more nuanced at an international level [27]. Its strength varied across *départements*, but accounting for deprivation and opportunistic screening rate patterns reduced this heterogeneity. The effect of travel time was stronger as age increased. We found no other publication studying this effect, thus further work is needed to appreciate its robustness. The number of radiology centres for 100,000 eligible women by *département* was not associated with screening uptake, in accordance with previous results [17]. Results are more nuanced at an international level [27]. Measure of specific care accessibility through this density may not be sufficient. More complete measures of accessibility (including social isolation, public transport availability, women’s travel possibilities or radiology centres characteristics) might be more informative. Our findings about EDI are in line with other studies using this index [15,16], Townsend index [17], or individual socioeconomic status [8,9,10,11,12]. The effect of deprivation on screening has also been reported multiple times, in multiple settings [27]. We did not find the “inverse U-curve” reported on the same screening campaign [18]. Multiple factors could explain this difference: the indexes used (FDEP [28] and EDI), their level of measure (municipalities and IRIS), the level of outcome assessment (municipality and individual) or differences in population (estimated and eligible–95 and 41 *départements*). However, we made the same observation that strength of social inequities varied across *départements*. Whereas participation among the most deprived was lower, but more comparable between *départements*, participation among the wealthiest was higher but more heterogeneous. We identified two factors explaining this heterogeneity. Firstly, participation to the FNBCSP was lower in *départements* with high opportunistic screening rates, mostly for the wealthiest. It has been reported that wealthiest populations tend to participate more to opportunistic screening [29,30,31]. Studies have also shown that opportunistic screening is more socially stratified than organized screening [32,33]. Moreover, it is most often a screening prescription by the general practitioner or gynaecologist [10,30], and France displays high income inequities in visiting both [8]. More aggressive breast tumours are more likely to be identified through their clinical manifestation, and could explain why some women could be led to opportunistic screening by the general practitioner or gynaecologist. We found no study in France linking tumour aggressiveness according to deprivation, but a study led in Denmark [34] showed that women with less education or lower income had higher risks of being diagnosed with a high-risk breast cancer. If this is also the case in France, we should observe the opposite effect (i.e., more participation in opportunistic screening as deprivation increases). Breast cancer incidence is higher for the wealthiest [35], which could justify a more intensive and individual follow-up for these women and their relatives. However, the size of this effect is low, and cannot explain such differences in opportunistic screening participation. Opportunistic screening, by competing with the FNBCSP, is thus one of the factors explaining low participation, and could hide a stronger social gradient in eligible population coverage. Secondly, participation was lower as *départements’* deprivation increased, here again with a stronger effect among the wealthiest. Although interesting, these results should be taken with caution as this is the first time that this measure was used—further work is needed in understanding deprivation patterns as well as robustness of these findings. The GDP-PPP was not associated with participation—this income-based socioeconomic status may not carry sufficient information to identify social inequities in FNBCSP participation. On note, neither social nor territorial inequities were found in a population having the possibility to choose between radiology centres and mobile mammography units [15]. In agreement with other publications [15,16,29,30,36], overall participation to the FNBCSP increased with age. Strength of this association varied across *départements*, with lower rates of screening uptake among the youngest in *départements* with higher opportunistic screening rates. This is consistent with other studies, showing that individual screening is mostly used by younger women [29,37], often starting before the recommended age range [36]. Lack of data on opportunistic screening leads to underestimating the true screening coverage, more importantly among the youngest. In the final model, the unexplained differences in FNBCSP participation between areas were reduced, mostly between *départements* (−82%) and to a lower extent between IRIS (−36%). We also explained most of the heterogeneities in the effects of age, travel time and EDI. Other studies are needed to find the factors influencing these remaining disparities. It should be noted that most of the heterogeneity in participation resided between individuals, with a total variance partition coefficient of 4% inside our levels (using the latent variable hypothesis).

This study has multiple strengths. First, with a representative sample of 42% of the estimated eligible population, this is the largest study about screening uptake using individual data led in France. In addition, collaboration with screening management structures allowed great accuracy in the study population. Even if, by default, homeless populations and women who recently moved are excluded, it can still be assumed that the invited population represents most of the eligible population. Additionally, *départements’* samples and populations did not differ greatly in age and participation to the FNBCSP. Although it does not guarantee representativeness, any major systematic differences would be unfortunate. High-precision geocoding for both women and radiology centres allowed to enrich screening database by adding travel time to the nearest accredited radiology centre and area-based deprivation measures. Finally, we tried to get the most out of multilevel models—which are particularly appropriate for nested data and contextual measures.

Some limitations also need to be addressed. By combining our 10% sample design and the unbalanced populations defined by the administrative boundaries of the IRIS, half of the IRIS had a population of less than 10 eligible women. Although it has been argued that the most important factor in multilevel analyses is the number of higher-level units [38], this could be of importance in estimating the model parameters. We excluded radiology centres from bordering *départements* because of poor data quality and did not account for opening and closing dates because of insufficient data. Radiology centres database needs to be created and updated by the public health authorities. We chose travel time from home to the nearest accredited radiology centre, but it is unknown whether it is the one chosen by eligible women, especially when accounting that a portion of them are professionally active. The data are not the most recent, but designing this study, the first of its kind, needed time (from recruiting screening management structures, getting ethical approval, acquisition of screening and radiology centre data, to geolocalisation, analyses and publication). However, updating the results on more recent campaigns using the same methodology could be done in a shorter time. The use of area-based deprivation index is known to raise the question of whether they act as a proxy for the individual status (e.g., people with high deprivation in these areas have their decision shifted towards no) and/or as contextual factors (e.g., living in an area with high deprivation shifts the decision of all its population towards no). It has been argued that adjusting for individual characteristics would help detangle these effects [39], but these data are not routinely collected. Finally, area-based deprivation is defined based on all population in these areas, while our population is age-gender-specific.

This work includes 41 *départements*. Even if screening organisation is the same in all French *départements*, our results may not be generalizable for all metropolitan France, especially for the *départements* of *Paris* and the *Hauts-de-Seine*, known to be the least participative in organized screening and in the most participative to opportunistic screening; in the island-*département* of *Corse*, with its particular geographical situation; and in the *département* of *Orne*, where a mobile mammographic unit is part of the screening programme. However, participation rates are very comparable between the *départements* in this study and the remaining French *départements* [5]. This study shows that it is logistically doable to develop a methodology which could be replicated to the whole metropolitan country. These results are also not generalizable to overseas *départements* for which the global context is quite different, and EDI might be inappropriate. Finally, this study evaluates only one screening campaign, and it is impossible to assert that every campaign suffers from the same pitfalls. However, our findings are in line with studies on other campaigns, and little has been done in targeting inequities in the FNBCSP. It is thus probable that inequities accumulate all along screening life, which could lead to a high loss of opportunities for some populations. In 2017, nearly all EU members had population-based BC screening programmes [40]. In a lot of them, coverage and participation remained low, and social inequities have often been reported, whether individual [41] or contextual [27,42]. Unfortunately, comparisons between these results and our findings are difficult because of the strong heterogeneity across measures and methodologies. EDI has been developed to be computable and comparable across all European countries and has already been developed in Spain, Portugal, Italy, Slovenia, and Lithuania. Our methodology is thus hypothetically replicable at the European level. The same approach could also be used in studying screening patterns for other cancer localisations.

## 5. Conclusions

BC is a particular disease in terms of social inequities—in France [35] and in Europe [43], its incidence (particularly for in situ cancers) is higher among the wealthiest populations. However, case fatality patterns follow the classical socioeconomic burden, with higher case fatality and shorter survival among the more unfavourable socioeconomic populations [44,45]. The populations that would benefit the most from a mortality reduction through screening are thus the most at risk of non-participation. This is in contradiction with two main goals of an organized screening programme: reducing mortality and ensuring equitable access. Additionally, the impact of the coexistence with opportunistic screening, beyond any control and evaluation, is adding more risks (over-diagnosis, overtreatment) [46], mainly in the youngest and wealthiest populations. Deeper evaluations are needed to evaluate the full implications of these results. The proportion of cancer found and missed, the stage at diagnosis, the follow-up to treatment, the effect on mortality, quality of life and the costs engendered and avoided by all these indicators need to be appreciated to allow an exhaustive evaluation. Research of immense value could be led with efficient linkage between cancer registries and screening databases. Some controversies about the benefit–risk balance of BC organized screenings have been widely discussed through the scientific community [47] and beyond [48]. They cannot be resolved without a better understanding of all consequences. A recent study at European level [49] estimated that yearly, 21,680 BC deaths were prevented due to mammography screening, and, with a hypothetical full coverage of 100%, 12,343 additional deaths could be prevented. Although this hypothetical coverage is not doable, the FNBCSP needs to evolve to be more efficient in both coverage of the population and reduction of the unfair gradient in participation.

## Figures and Tables

**Figure 1 cancers-13-04374-f001:**
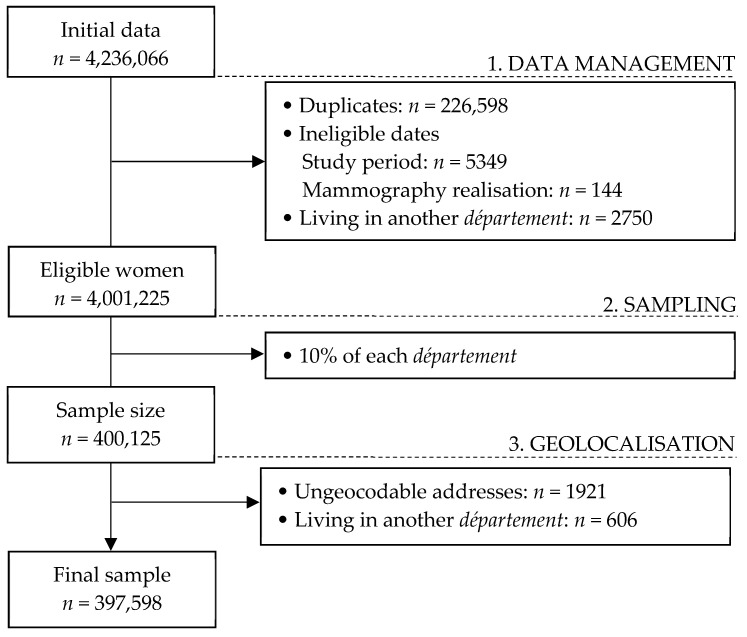
Flow chart of the population.

**Figure 2 cancers-13-04374-f002:**
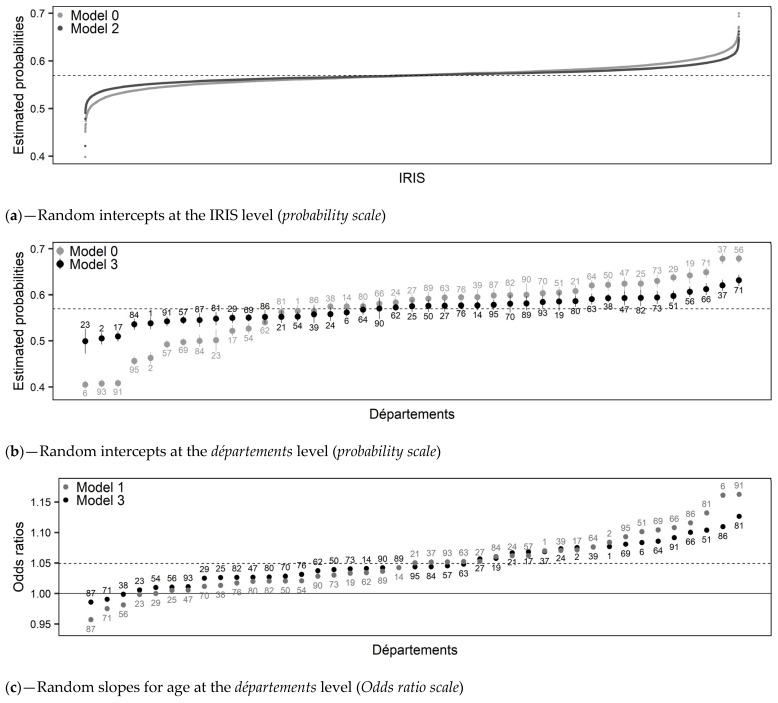
Illustrations of the random effects. (**a**) Random intercepts at the IRIS level (*probability scale*), (**b**) Random intercepts at the *départements* level (*probability scale*), (**c**) Random slopes for age at the *départements* level (*Odds ratio scale*), (**d**) Random slopes for travel time at the *départements* level (*Odds ratio scale*), (**e**) Random slopes for EDI at the *départements* level (*Odds ratio scale*).

**Figure 3 cancers-13-04374-f003:**
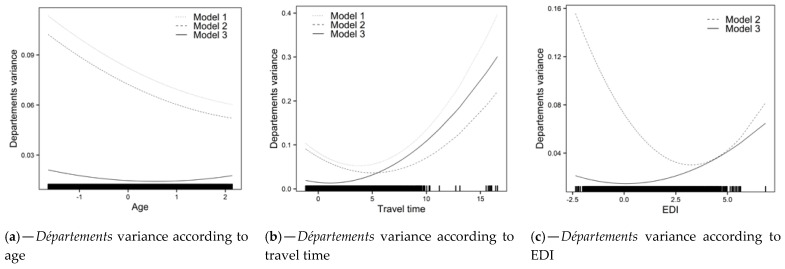
Variance between *départements* according to lower-level variables. (**a**) *Départements* variance according to age, (**b**) *Départements* variance according to travel time, (**c**) *Départements* variance according to EDI.

**Figure 4 cancers-13-04374-f004:**
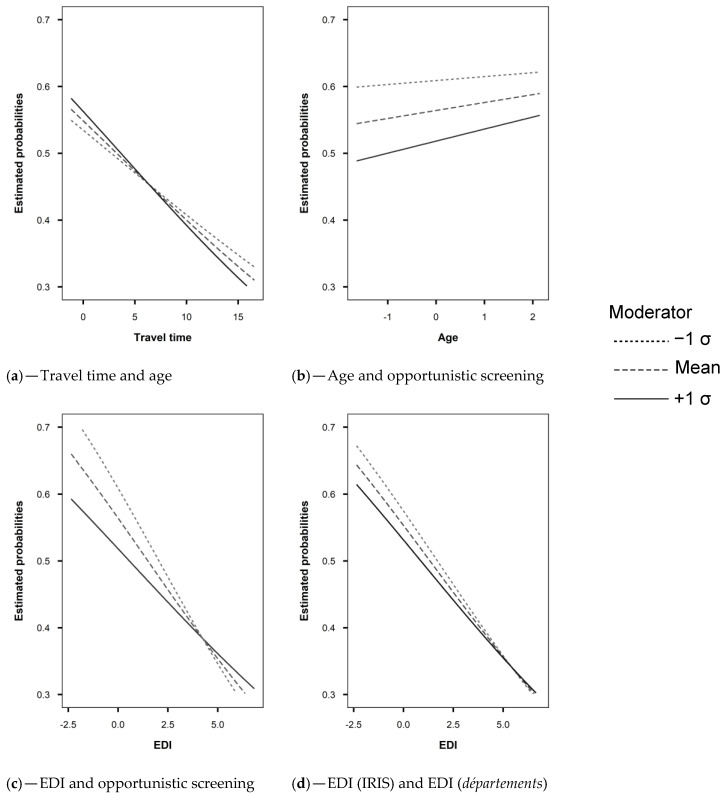
Interactions in the final model. (**a**) Travel time and age, (**b**) Age and opportunistic screening, (**c**) EDI and opportunistic screening, (**d**) EDI (IRIS) and EDI (*départements*).

**Table 1 cancers-13-04374-t001:** Characteristics of the population.

Level 1—Individual Level	Mean	Standard Deviation	Minimum	Maximum
Travel time (minutes)	8.70	7.47	0.00	132.48
Age (years)	60.73	7.11	50	74
FNBCSP ^a^ participation (%)	55.06	/	/	/
**Level 2—IRIS ^b^**				
EDI ^c,d^ (dimensionless)	0.97	5.12	−11.08	35.96
**Level 3—*Départements***				
Opportunistic screening rates (%)	8.91	6.06	2.30	28.00
Mean of EDI ^c,d^ (dimensionless)	0.97	2.29	−1.67	9.63
GDP (PPP) ^e^ per capita (US$)	20,638	6134	17,310	23,360
Number of accredited radiology centres/100,000 eligible women	21.88	8.55	7.69	59.06

^a^. French national breast cancer screening programme, ^b^. Îlots regroupés pour l’information statistique, ^c^. European Deprivation Index, ^d^. Population weighted, ^e^. Gross-domestic product based on purchasing power parity.

**Table 2 cancers-13-04374-t002:** Individual and contextual factors associated with French National Breast Cancer Screening Programme participation.

	Model 0: Empty Model	Model 1: Level 1 Variables	Model 2: Level 2 Variable	Model 3: Level 3 Variables
**Level 1—Individuals**				
Intercept	1.32 [1.22–1.45]	1.35 [1.24–1.47]	1.32 [1.22–1.43]	1.32 [1.27–1.37]
Age	/	1.05 [1.03–1.07]	1.05 [1.03–1.07]	1.05 [1.04–1.06]
Travel time	/	0.98 [0.96–0.99]	0.95 [0.93–0.96]	0.94 [0.93–0.95]
Age × travel time	/	0.99 [0.98–1.00]	0.99 [0.98–1.00]	0.99 [0.98–1.00]
**Level 2—IRIS**				
*Fixed effects*				
EDI	/	/	0.84 [0.82–0.86]	0.84 [0.83–0.85]
*Random effects*				
Random intercept (σ^2^_0I_)	0.055 [0.048–0.058]	0.053 [0.048–0.058]	0.035 [0.030–0.039]	0.035 [0.031–0.039]
VCP	1.60%	1.55%	1.03%	1.05%
PCV (compared with empty model)	/	−3.64%	−36.36%	−36.36%
**Level 3—*Départements***				
*Fixed effects*				
Individual screening rates	/	/	/	0.84 [0.79–0.87]
Deprivation	/	/	/	0.91 [0.88–0.96]
*Cross-level interactions*				
Individual screening rates × Age	/	/	/	1.02 [1.01–1.04]
Individual screening rates × EDI	/	/	/	1.04 [1.03–1.06]
Mean of EDI × EDI	/	/	/	1.02 [1.00–1.03]
*Random effects*				
Random intercept (σ^2^_0D_)	0.082 [0.053–0.130]	0.082 [0.048–0.123]	0.072 [0.044–0.108]	0.015 [0.007–0.021]
VPC	2.39%	2.39%	2.12%	0.45%
PCV (compared with empty model)	/	0%	−12.20%	−81.71%
Age random slope (σ^2^_1D_)	/	2.3 × 10^−3^ [1.2 × 10^−3^–3.7 × 10^−3^]	2.3 × 10^−3^ [1.2 × 10^−3^–3.6 × 10^−3^]	1.4 × 10^−3^ [5.3 × 10^−4^–2.2 × 10^−3^]
Travel time random slope (σ^2^_2D_)	/	2.1 × 10^−3^ [1.0 × 10^−3^–3.4 × 10^−3^]	1.4 × 10^−3^ [5.3 × 10^−4^–2.3 × 10^−3^]	1.2 × 10^−3^ [4.0 × 10^−4^–2.2 × 10^−3^]
EDI random slope (σ^2^_3D_)	/	/	4.1 × 10^−3^ [1.8 × 10^−3^–6.7 × 10^−3^]	1.1 × 10^−3^ [1.0 × 10^−4^–1.8 × 10^−3^]
*Random effects correlation*				
σ^2^_0D,_ σ^2^_1D_	/	−0.55 [−0.77; −0.19]	−0.55 [−0.78; −0.23]	−0.18 [−0.57; 0.23]
σ^2^_0D,_ σ^2^_2D_	/	−0.60 [−0.83; −0.31]	−0.71 [−0.94; −0.42]	−0.32 [−0.67; 0.13]
σ^2^_0D,_ σ^2^_3D_	/	/	−0.76 [−0.91; −0.54]	−0.03 [−0.65; 0.59]
σ^2^_1D,_ σ^2^_2D_	/	0.49 [0.12; 0.82]	0.68 [0.34; 0.95]	0.55 [0.10; 0.93]
σ^2^_1D,_ σ^2^_3D_	/	/	0.43 [0.08; 0.75]	−0.04 [−0.72; 0.70]
σ^2^_2D,_ σ^2^_3D_	/	/	0.60 [0.17; 0.89]	−0.09 [−0.87; 0.52]
**Deviance**	536,474	535,848	534,615	534,549

## Data Availability

Access to the data that support the findings of this study is restricted. These data are not publicly available.

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
