# Peer review of "Socio-Territorial Inequities in the French National Breast Cancer Screening Programme—A Cross-Sectional Multilevel Study"

_cancers, 2021, doi:10.3390/cancers13174374_

Round 1

Reviewer 1 Report

Very nice analysis. Although 'opportunistic screening' was an independent variable in the multi-level modeling approach, the authors may consider devoting greater attention to describing 'opportunistic screening' for higher EDI. As mentioned, 'opportunistic screening' is a crude variable that includes physician referrals following clinical anomalies. Aggressive breast tumors are less likely to be picked up within a formal screening program; patients with these tumors are likely to experience clinical manifestations of the disease that prompts them to see a physician who refers them for a mammogram. If economically deprived/privileged persons are engaged in opportunistic screening, it may be due to population subgroups experiencing a higher incidence of aggressive tumors. Thus, the discussion could include a section on incidence of molecular subtyped breast tumors in France - or, specifically, across the 41 departments in the study, if such data is available.

Author Response

Response to Reviewer 1 Comments

Point 1 – Although 'opportunistic screening' was an independent variable in the multi-level modeling approach, the authors may consider devoting greater attention to describing 'opportunistic screening' for higher EDI. As mentioned, 'opportunistic screening' is a crude variable that includes physician referrals following clinical anomalies. Aggressive breast tumors are less likely to be picked up within a formal screening program; patients with these tumors are likely to experience clinical manifestations of the disease that prompts them to see a physician who refers them for a mammogram. If economically deprived/privileged persons are engaged in opportunistic screening, it may be due to population subgroups experiencing a higher incidence of aggressive tumors. Thus, the discussion could include a section on incidence of molecular subtyped breast tumors in France - or, specifically, across the 41 departments in the study, if such data is available.

Response 1 – Unfortunately, this is actually impossible to compute these data for France. These data could be available in the French cancer registries, covering only 22 French départements ­– and they are not linked to screening databases. Yet, this is an interesting point, and we have tried to give more elements in the discussion:

“More aggressive breast tumours are more likely to be identified through their clinical manifestation, and could explain why some women could be led to opportunistic screening by the general practitioner or gynaecologist. We found no study in France linking tumour aggressiveness according to deprivation, but a study led in Denmark [34] showed that women with less education or lower income had higher risks of being diagnosed with a high-risk breast cancer. If this is also the case in France, we should observe the opposite effect (i.e. more participation in opportunistic screening as deprivation increases). Breast cancer incidence is higher for the wealthiest [35], which could justify a more intensive and individual follow-up for these women and their relatives. However, the size of this effect is low, and cannot explain such differences in opportunistic screening participation.”

Point 2. English language and style are fine/minor spell check required

Response 2. The rewrote manuscript went under revision by a native english speaker.

Reviewer 2 Report

This article deals with disparities in access to routine breast cancer screening implemented in France in 2004.

Because the data are from 2013 and 2014, the article can be seen as an assessment after 10 years of implementation of the program. This approach should be mentioned in the article. The use of data that are a bit old should be justified.

The article is synthetic and clear. However, before publication, some improvements are necessary.

Main points:

  • Only 41 French departments were included in the study out of 101. This point is noted and discussed but the methodological choice is not justified. How is the study population representative of the French population of women expected to access to the screening? How were the departments selected? What was the proportion of the population of women eligible for screening included? Unless I am mistaken, only some departments in Paris region (French main city) were included. What is the reason for this choice?
  • It would be interesting to add a column in Table 1 mentioning the same characteristics but for the whole French population as comparison.
  • Figure 2 is very difficult to understand. Instead of b, c, d, e in the graphs, it would be better to specify what the Odds Ratios correspond to.

Author Response

Response to Reviewer 2 Comments

Point 1. Because the data are from 2013 and 2014, the article can be seen as an assessment after 10 years of implementation of the program. This approach should be mentioned in the article. The use of data that are a bit old should be justified. The article is synthetic and clear. However, before publication, some improvements are necessary.

Response 1. We have added a point about this in the limitation of this study:

“The data are not the most recent, but designing this study, the first of its kind, needed time (from recruiting screening management structures, getting ethical approval, acquisition of screening and radiology centre data, to geolocalisation, analyses and publication). However, updating the results on more recent campaigns using the same methodology could be done in a shorter time.”

Point 2. Only 41 French departments were included in the study out of 101. This point is noted and discussed but the methodological choice is not justified. How is the study population representative of the French population of women expected to access to the screening? How were the departments selected? What was the proportion of the population of women eligible for screening included? Unless I am mistaken, only some departments in Paris region (French main city) were included. What is the reason for this choice?

Response 2 – We have added multiple elements to answer this point:

In material and method:

  • “Participation of the structures was on a voluntary basis.”
  • “Data management was performed, leaving 4 001 225 (94%) unique eligible women, 42% of the estimated eligible population in France”

In the discussion:

  • “This study has multiple strengths. First, with a representative sample of 42% of the estimated eligible population, this is the largest study about screening uptake using individual data led in France”
  • “This work includes 41 départements. Even if screening organisation is the same in all French départements, our results may not be generalizable for all metropolitan France, especially for the départements of Paris and the Hauts-de-Seine, known to be the least participative in organized screening and in the most participative to opportunistic screening; in the island-département of Corse, with its particular geographical situation; and in the département of Orne, where a mobile mammographic unit is part of the screening programme. However, participation rates are very comparable between the départements in this study and the remaining French départements [5]

Point 3. It would be interesting to add a column in Table 1 mentioning the same characteristics but for the whole French population as comparison.

Response 3. Unfortunately, the characteristics described are not available in the whole screening eligible French population. Age needed receiving the data, travel time, number of accredited radiology centres and EDI also needed geolocalisation, and participation rate to the FNBCSP is uncalculated at the individual level in France.

Point 4. Figure 2 is very difficult to understand. Instead of b, c, d, e in the graphs, it would be better to specify what the Odds Ratios correspond to.

Response 4. We have accounted for this remark in the figure 2, and in all other figures.

Reviewer 3 Report

Technical Notes: In the abstarct, the incomprehensible abbreviation (IRIS) is used; some drawings do not have the axes described (Fig. 4).

line 32-35 - information on deaths is incorrect or imprecise. Breast cancer is the most common cancer among women and continues to be the most common cause of cancer death in the world.

In my opinion, "Introduction" is extremely detailed, I would recommend shortening it, especially since the reader does not know where Calavados or Orme is. The reader also does not know how these regions are representative or not for the rest of the country.

Population and sample - I recommend shortening the description, everything is very well explained in the diagram. In addition, I would recommend checking how the analyzed sample is consistent with the general population.

The models used confirm the intuitive knowledge that higher EDI and more difficult access to medical examination translates into lower participation. The limitation of the study is that it relates the results to the studied population only, although perhaps it could be compared with other European countries

Author Response

Response to Reviewer 3 Comments

Point 1. In the abstract, the incomprehensible abbreviation (IRIS) is used

Response 1. We have added the information

“These observations were nested within 22 250 residential areas called “Îlots Regroupés pour l'Information Statistique” (IRIS), for which the European Deprivation Index (EDI) is defined.”

Point 2. Some drawings do not have the axes described (Fig. 4).

Response 2. We have described the axes where it was missing.

Point 3. line 32-35 - information on deaths is incorrect or imprecise. Breast cancer is the most common cancer among women and continues to be the most common cause of cancer death in the world.

Response 3. Our formulation might have been misleading; this was the presentation “all sexes combined”. We have added some precisions:

“Worldwide, one in six deaths is due to cancer. Breast cancer (BC), despite being mostly a female disease (less than 1% occurs in men), has now surpassed lung cancer as the most frequently diagnosed cancer, all sexes combined, with 2.3 million cases in 2020. With 685 000 deaths the same year, it ranked fifth for the most common cause of cancer death all sexes combined, and first in women [1]. In France, after a stabilization between 2003 and 2010, incidence has increased again during 2010-2018. In contrast, mortality slowly decreased between 1990 and 2018. Therefore, in 2018, BC was the most diagnosed cancer (58 459 new cases) and ranked third in mortality all sexes combined, and first in women (12 146 deaths) [2].”

Point 4. In my opinion, "Introduction" is extremely detailed, I would recommend shortening it, especially since the reader does not know where Calvados or Orme is. The reader also does not know how these regions are representative or not for the rest of the country.

Response 4. We have shortened this part:

“Higher deprivation was associated with lower participation in the FNBCSP in two départements (over 101) [15,17], and in a representative sample of the eligible population covered by the three main health insurance schemes in thirteen départements [16].”

Point 5. Population and sample - I recommend shortening the description, everything is very well explained in the diagram.

Response 5. We have shortened this part:

“Screening management structures were invited to send their data regarding the 2013-2014 invitation campaign. Participation of the structures was on a voluntary basis. These data corresponded to the follow-up of the FNBCSP and included eligible women’s addresses of residence, dates of birth, dates of invitation (from which we computed age at invitation), and whether they performed a mammography screening through the FNBCSP in the two years following invitation. We received 4 236 066 observations nested in 41 départements. Data management was performed, leaving 4 001 225 (94%) unique eligible women, 42% of the estimated eligible population in France. Before geolocalisation, we performed a stratified random sampling by drawing 10% of the eligible population in each département (n = 400 125). Comparisons between samples and départements’ populations (data not shown) showed no important differences in participation rates (from 0 to 1.6% difference) and age (no differences exceeded ¼ years in mean age). Additional exclusions were performed due to geolocalisation and after geolocalisation. Final sample consisted of 397 598 women. Flow chart of the population is available in Figure 1. “

Point 6. In addition, I would recommend checking how the analyzed sample is consistent with the general population

Response 6. Unfortunately, the characteristics described are not available in the whole screening eligible French population. We have added information on the representativeness of our sample:

In material and method:

  • “Participation of the structures was on a voluntary basis.”
  • “Data management was performed, leaving 4 001 225 (94%) unique eligible women, 42% of the estimated eligible population in France”

In the discussion:

  • “This study has multiple strengths. First, with a representative sample of 42% of the estimated eligible population, this is the largest study about screening uptake using individual data led in France”
  • “This work includes 41 départements. Even if screening organisation is the same in all French départements, our results may not be generalizable for all metropolitan France, especially for the départements of Paris and the Hauts-de-Seine, known to be the least participative in organized screening and in the most participative to opportunistic screening; in the island-département of Corse, with its particular geographical situation; and in the département of Orne, where a mobile mammographic unit is part of the screening programme. However, participation rates are very comparable between the départements in this study and the remaining French départements [5]